# Scoring protein-ligand binding structures through learning atomic graphs with inter-molecular adjacency

**Debby D. Wang**[1]*, **Yuting Huang**[1]

School of Science and Technology, Hong Kong Metropolitan University, Ho Man Tin, Hong Kong

* dwang@hkmu.edu.hk

## Abstract

With a burgeoning number of artificial intelligence (AI) applications in various fields, biomolecular science has also given a big welcome to advanced AI techniques in recent years. In this broad field, scoring a protein-ligand binding structure to output the binding strength is a crucial problem that heavily relates to computational drug discovery. Aiming at this problem, we have proposed an efficient scoring framework using deep learning techniques. This framework describes a binding structure by a high-resolution atomic graph, places a focus on the inter-molecular interactions and learns the graph in a rational way. For a protein-ligand binding complex, the generated atomic graph reserves key information of the atoms (as graph nodes), and focuses on inter-molecular interactions (as graph edges) that can be identified by introducing multiple distance ranges to the atom pairs within the binding area. To provide more confidence in the predicted binding strengths, we have interpreted the deep learning model from the model level and in a post-hoc analysis. The proposed learning framework has been demonstrated to have competitive performance in scoring and screening tasks, which will prospectively promote the development of related fields further.

**Data availability statement:** All the original data for model construction are available from the PDBbind database (https://www.pdbbind-plus.org.cn/). Specifically, the Version 2020 (PDBbind v2020)

## Author summary

The binding between a small compound (ligand) and a protein plays a crucial role in many biological processes, such as signal transduction and immunoreaction. Particularly, a small-molecule drug can bind to a target protein to modulate its signaling pathways and suppress the progression of the associated disease. Apparently, the binding strength is a key indicator for evaluating how well such small-molecule drugs work, therefore becoming a core topic in computational drug discovery. Nowadays, the binding structure of a ligand and its target protein can be resolved experimentally or modeled computationally, while the accurate scoring of such a binding structure (predicting the binding strength) still remains a challenge. An effort has been put into the development of benchmark databases that provide a variety of protein-ligand binding structures and

was used in this work, and it can be accessed from the Download section of PDBbind+ website. The CASR database was used for evaluating the scoring performance of constructed models (https://www.ncbi.nlm.nih.gov/pmc/articles/PMC3753885/). These data have been cleaned, standardized, and stored in Zenodo (https://zenodo.org/records/15023336). The screening power of the constructed models were measured using the data from DUD-E (https://dude.docking.org/). All code files are available from an online GitHub repository at https://github.com/debbydanwang/DL-PLBAP. A Docker container with a trained model pre-installed is available for access on Zenodo (https://zenodo.org/records/15023336).

**Funding:** This work was supported by Hong Kong Research Grants Council (Project UGC/FDS16/E16/23 to DDW) and Hong Kong Metropolitan University (Project 2023/24 S&T to DDW). The funders had no role in study design, data collection and analysis, decision to publish, or preparation of the manuscript. The authors received salaries from Hong Kong Metropolitan University.

**Competing interests:** The authors have declared that no competing interests exist.

their experimentally resolved binding strengths, leading to increasing deep learning applications in this field. In this study, we represent a protein-ligand binding structure as a graph, with the atoms as nodes and the inter-molecular interactions as edges. A light but efficient deep learning architecture has been adopted for learning such graphs and outputting the binding strengths. Validated by our experiments, the model performs well in both scoring and screening tasks.

## Introduction

'AI for science' has attracted considerate attention in the past decade. Quite a number of powerful mathematical algorithms have been developed in this field, to rise to the challenging tasks in dermatology [1], precision medicine [2], molecular science [3] and drug discovery [4].

As a crucial problem in computer-aided drug discovery (CADD), scoring a protein-ligand complex structure to exhibit its binding strength (Fig 1) always seeks for breakthroughs in AI developments. Such binding strength, as a key indicator of the efficacy of a drug that attaches to its target protein, can mostly be attributed to various non-covalent interactions (e.g. hydrogen bonds, hydrophobic contacts and $\pi$-stacking). Earlier AI-based scoring works leveraged traditional machine-learning algorithms (e.g. random forests) to parse a feature vector, which describes the interactions in a protein-ligand complex structure, and mapped the vector to the binding strength [5–10]. It lasts until the emergence of deep learning, which reached its scientific milestones by the launch of *AlphaFold* (for near-perfect protein-fold predictions) [11] and the *GPT*-series (strong human-like chatbots) [12].

When first introduced to the works of scoring molecular binding strength, deep learning was primarily utilized in the manner of convolutional neural networks (CNNs) [13–18]. Accordingly, molecular lattices or grids, with each cell characterized by a collection of atomic properties (e.g. physico-chemical or pharmacophoric), are the *de facto* feature representations of a protein-ligand complex. The KDEEP model adopts a molecular lattice representation with a size of 24Å × 24Å × 24Å and a set of eight atomic properties (pharmacophoric) for delineating a complex structure, and feeds the lattice into a 3D-CNN for binding strength prediction [13]. The Pafnucy model compresses 19 atomic properties (both physico-chemical and

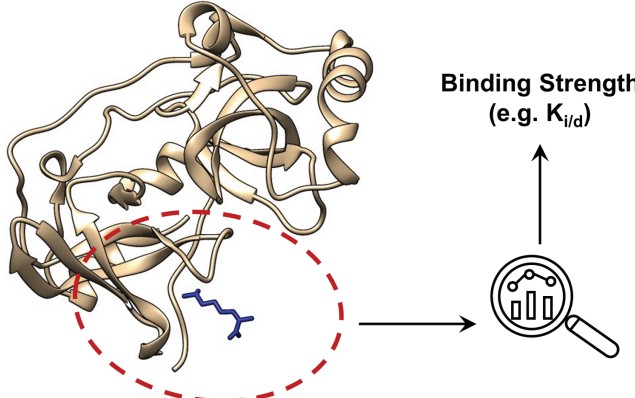

**Fig 1. Scoring problem.** Scoring a protein-ligand complex structure to exhibit the binding strength.

pharmacophoric) of a complex structure into a molecular lattice ($20\text{Å} \times 20\text{Å} \times 20\text{Å}$), and employs a simple 3D-CNN architecture for learning the lattice [14]. Rezaei et al. developed a light-weight 3D-CNN model for scoring, based on $32\text{Å} \times 32\text{Å} \times 32\text{Å}$ molecular lattices that concern 24 atomic features (11 Arpeggio atom types and the excluded volume for both protein and ligand) [15]. Although has opened a new venue for scoring works, deep lattice learning often lacks rotational invariance in data and is therefore resource-intensive after data augmentation [13,14].

More recently, molecular graph learning has become a prevalent technique for the scoring works. In this context, a protein-ligand complex structure is commonly represented as a 2D atomic graph, which is then decoded by graph neural networks (GNNs) [19–23]. GraphBAR adopts a molecular graph with distance-dependent edges to characterize the binding-site atoms in a complex structure, and employs a spectral graph convolutional network (GCN) to map the graphs to the binding strengths [20]. Shen et al. considered the covalent connections for atoms in the binding area, and leveraged a cascade GCN (with two concatenated modules) for graph learning and binding strength prediction [21]. GraphscoreDTA represents a complex by a fusion of graphs (a 1D amino-acid graph for protein, an atomic graph for ligand, and a hybrid graph for the binding pocket), and predicts the binding strength using a GNN with a bitransport information mechanism and Vina distance terms [23]. Zhang et al. utilized a similar graph representation and developed a multi-objective GNN model for binding-strength scoring [22]. These pioneer works have shed light on modern scoring works. Nevertheless, there is still much room for improvement in developing target-oriented graph representation, achieving high screening power and making the model more transparent. Accordingly, we are dedicated to the design of efficient scoring models, with informative molecular graphs, descent screening power and reasonable interpretability, in this work.

## Materials and methods

### Atomic-level molecular graphs

A molecular graph can be represented as $\mathcal{G} = (\mathbf{V}, \mathbf{E})$, where $\mathbf{V}$ indicates nodes $\{nd_1^0, \dots, nd_{|V|}^0\}$ and $\mathbf{E}$ stands for edges connecting those nodes $\{eg_1^0, \dots, eg_{|E|}^0\}$.

To capture sufficient information in a molecule, treating its atoms as graph nodes is a well-acknowledged strategy. Each node or atom is then characterized by a series of physico-chemical or pharmacophoric properties, leading to a feature matrix $\mathbf{F}^0 \in \mathbb{R}^{|V| \times m}$ of all the nodes in the graph ($m$ is the number of properties). As molecules like proteins are very large in terms of atoms, retaining all the atoms is a heavy burden to the computations and therefore task-oriented cropping is frequently performed. For scoring a protein-ligand binding structure, the atoms in the binding area is often of interest. This results in a smaller feature matrix $\mathbf{F} \in \mathbb{R}^{n \times m}$, where $n$ is the number of nodes in the binding area ($\mathbf{V}^{ba} = \{nd_1, \dots, nd_n\}$).

Generally, a graph in deep learning works shows the connections between nodes by an adjacency matrix $\mathbf{A}$, where $A_{ij}$ indicates an edge between the $i$-th and $j$-th nodes. However, designing task-specific graph edges, especially for tasks involving molecules, is often challenging. The covalent bonds, contacts defined through distance thresholding, or a combination of them have been regarded as edges in different works [20,21,24]. Considering the atoms in the binding area of a protein-ligand complex, Fig 2A shows the covalent adjacency among those atoms. Interactions or contacts between a pair of atoms ($nd_i$ and $nd_j$) can also be defined by the range where the atomic distance ($d_{ij}$) resides, leading to multi-level distance-dependent adjacencies among atoms. Fig 2B displays two types of atomic contacts ($d_{ij} \leq 3\text{Å}$ and $3\text{Å} < d_{ij} \leq 4\text{Å}$). In Fig 2C, a hybrid type of adjacencies (covalent bond and distance-dependent

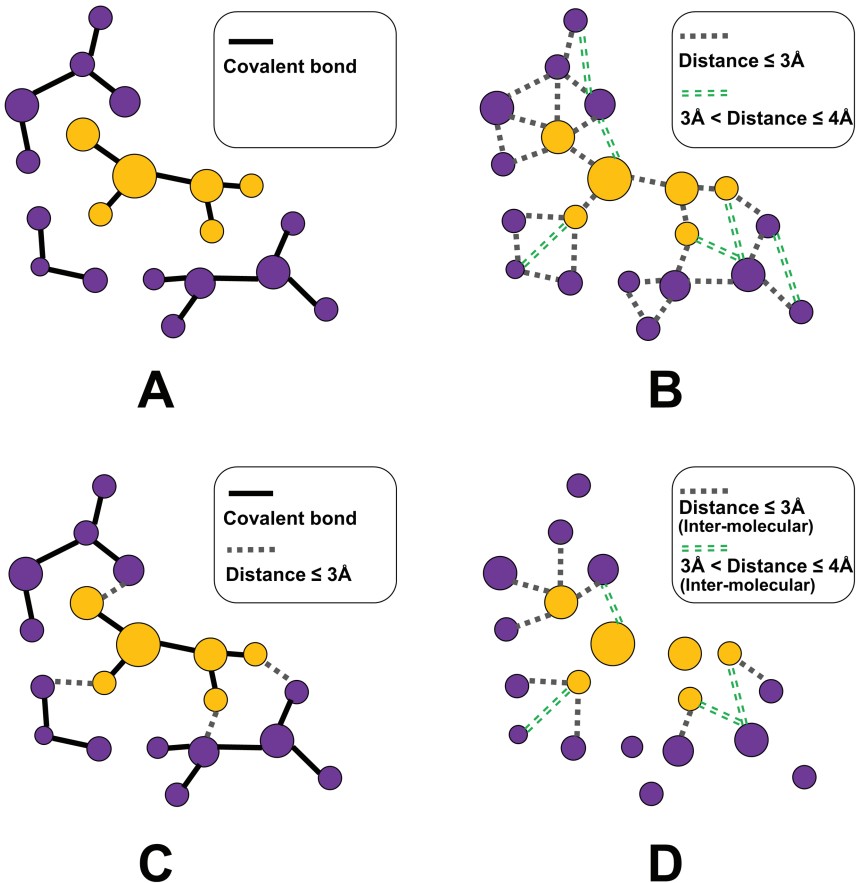

**Fig 2. Different definitions of atomic adjacency in a molecular graph. A**. Covalent adjacency. **B**. Distance-dependent contacts. **C**. A combination of covalent adjacency and distance-dependent contacts. **D**. Inter-molecular contacts through distance thresholding.

contacts) is considered. *For scoring tasks, these adjacency definitions either emphasize the covalent bonds, or mix the inter- and intra-molecular interactions, resulting in the loss of focus on the inter-molecular interactions.* Nevertheless, these inter-molecular interactions play a pivotal role in determining the binding strength between a ligand and its target protein. Accordingly, we focus on the inter-molecular contacts in this work, and define multi-level atomic adjacencies by one-hot encoding of those contacts belonging to different distance ranges (Fig 2D). Such adjacencies can be stored in an adjacency tensor $\mathbf{A}$, where each slice $\mathbf{A}_{::k}$ shows all pairs of nodes having distances in range $\delta_k$, as follows.

$$A_{ijk} = \begin{cases} 1 & nd_i \text{ and } nd_j \text{ is a protein-ligand atom pair} \\ & \& \ d_{ij} \in \delta_k \\ 0 & otherwise \end{cases} \tag{1}$$

Algorithm 1 shows the procedure for generating such an adjacency tensor for a protein-ligand binding area.

**Algorithm 1** Generating an Inter-molecular Adjacency Tensor

```
Input: Coordinates Coord₁,…,Coordₙ for the atoms in the binding area
(nd₁,…,ndₙ), a list of distance ranges δ₁,…,δₖ
Output: An inter-molecular adjacency tensor A
Initialize A = 0 (∈ ℝⁿˣⁿˣᴷ).
Calculate the distance matrix D ∈ ℝⁿˣⁿ based on Coord₁,…,Coordₙ.
for k = 1 to K do
    A::k = I_{D∈δₖ}                              ▷ I_{D∈δₖ} is an indicator function
    for i = 1 to n do
        for j = i+1 to n do
            if A_{ijk} ≠ 0 and ndᵢ-nd_j is not a protein-ligand atom pair then
                A_{ijk} = A_{jik} = 0            ▷ Turn off intra-molecular interactions
            end if
        end for
    end for
end for
```

## Graph-based deep learning

Given a graph with a node-feature matrix $\mathbf{F}$ and an adjacency matrix $\mathbf{A}$, message-passing mechanisms are frequently adopted for learning such a graph [25]. These mechanisms, as shown in Eq 2, update nodes features iteratively in a local context.

$$\mathbf{f}_i^{l+1} = \alpha_l(\mathbf{f}_i^l, \bigoplus_{j\in\mathcal{N}(i)} \mathbf{m}_{j\rightarrow i}^l) \tag{2}$$

Here, $\mathbf{f}_i^l$ is the features describing the $i$-th node in the $l$-th layer, $\mathcal{N}(i)$ indicates the neighborhood of the $i$-th node (based on the adjacency matrix), $\mathbf{m}_{j\rightarrow i}^l$ is the message passed from the $j$-th node to the $i$-th node in the $l$-th layer, $\bigoplus$ denotes a permutation-invariant function (e.g. average), and $\alpha_l$ is an update function such as a neural network.

Although a wide variety of graph neural networks (GNNs) have been developed, properly learning molecular graphs still remains a challenge. The *ChebNet* [26], leveraging spectral graph convolutions, is among the well-acknowledged GNNs. It has an efficient form for updating node features in each iteration, as follows.

$$\mathbf{F}^{l+1} = \sigma_l(\tilde{\mathbf{D}}^{-\frac{1}{2}}\tilde{\mathbf{A}}\tilde{\mathbf{D}}^{-\frac{1}{2}}\mathbf{F}^l\mathbf{W}^l) \tag{3}$$

Here, $\mathbf{F}^l$ is the feature matrix for all the nodes in the $l$-th layer, $\sigma_l$ is an activation function, $\mathbf{W}^l$ is the weight matrix, and $\tilde{\mathbf{D}}^{-\frac{1}{2}}\tilde{\mathbf{A}}\tilde{\mathbf{D}}^{-\frac{1}{2}}$ is a normalized adjacency matrix with self-adjacencies ($\tilde{\mathbf{A}} = \mathbf{A} + \mathbf{I}_n$ and $\tilde{D}_{ii} = \sum_j \tilde{A}_{ij}$). Such graph-learning operations can be stacked into $L$ layers. From the message-passing perspective, this mechanism can be regarded as a simple average of the normalized information collected from the neighborhood of a node.

$$\mathbf{f}_i^{l+1} = \alpha_l(\frac{1}{\tilde{D}_{ii}}\mathbf{f}_i^l\mathbf{W}^l + \sum_{j\in\mathcal{N}(i)} \frac{\tilde{A}_{ij}}{\sqrt{\tilde{D}_{ii}\tilde{D}_{jj}}}\mathbf{f}_j^l\mathbf{W}^l) \tag{4}$$

Previously, *ChebNet* has been questioned about its capability for capturing long-range dependence among the nodes in a graph. *However, scoring protein-ligand binding strength is a work that largely concerns local contexts (e.g. a key hydrogen bond or an important interaction), making the ChebNet mechanism fit well in this task.* When focusing on only the inter-molecular interactions (Fig 2D), we update the features once (Eq 3) to learn the

neighborhoods of binding-site atoms in this work. Higher-order graph convolutions, which will involve intra-molecular interactions and be computationally expensive, are not considered. This strategy places an absolute focus on the inter-molecular interactions (crucial to scoring works) and is of high efficiency. Since $L = 1$ in this scenario, the layer notation $l$ will be omitted for simplicity in what follows.

Instead of using a single adjacency matrix $\mathbf{A}$, an adjacency tensor that covers different adjacency (interaction) types is of necessity in a scoring task. As inter-molecular interactions are mostly non-covalent (atomic distance $d_{ij} > 2\mathring{A}$), multiple distance ranges starting from $2\mathring{A}$ can be nominated to construct the inter-molecular adjacency tensor. Fig 2D exhibits a two-slice adjacency tensor $\mathbf{A}$ as follows.

$$
\begin{aligned}
A_{ij1} &= \begin{cases} 1 & nd_i \text{ and } nd_j \text{ is a protein-ligand atom pair} \\ & \& \ 2\mathring{A} < d_{ij} \leq 3\mathring{A} \\ 0 & otherwise \end{cases} \\[2ex]
A_{ij2} &= \begin{cases} 1 & nd_i \text{ and } nd_j \text{ is a protein-ligand atom pair} \\ & \& \ 3\mathring{A} < d_{ij} \leq 4\mathring{A} \\ 0 & otherwise \end{cases}
\end{aligned}
\tag{5}
$$

Here we adopt such a tensor because $4\mathring{A}$ has been verified to be a distance threshold for capturing sufficient inter-molecular interactions in a binding complex [20].

Targeting at each type of inter-molecular contacts ($k = 1, 2$), the graph nodes can be learned using the message-passing mechanism in *ChebNet*, as

$$
\mathbf{F}^k = \sigma\big(\tilde{\mathbf{D}}_k^{-\frac{1}{2}} \tilde{\mathbf{C}}_k \tilde{\mathbf{D}}_k^{-\frac{1}{2}} \mathbf{F} \mathbf{W}_k\big)
\tag{6}
$$

where $\tilde{\mathbf{C}}_k = \mathbf{A}_{::k} + \mathbf{I}_n$, $\tilde{\mathbf{D}}_k$ is a diagonal matrix showing the degree of each node in $\tilde{\mathbf{C}}_k$, and all the other notations follow Eq 3.

After collecting the messages from direct neighbors of graph nodes, we gather the information into the graph level for the scoring purpose. Such an aggregation function is permutation-invariant and similar to that in Eq 2. A simple summation in the following equation serves as an example.

$$
\mathbf{h}_k = \sum_{i \in \mathbf{V}^{ba}} \mathbf{f}_i^k
\tag{7}
$$

The features for different inter-molecular interactions ($k = 1, 2$) are then concatenated before being fed into dense layers for final graph-level predictions.

$$
\mathbf{h} = \|_k \mathbf{h}_k
\tag{8}
$$

Here, $\|$ indicates an concatenation of features and $\mathbf{h}$ stands for the hidden features describing the whole binding-site graph.

Referring to a well-established architecture (*GraphBAR* [20]), we developed a light graph-learning architecture that focuses only on inter-molecular interactions and learns the interactions through direct atomic neighborhoods (Fig 3).

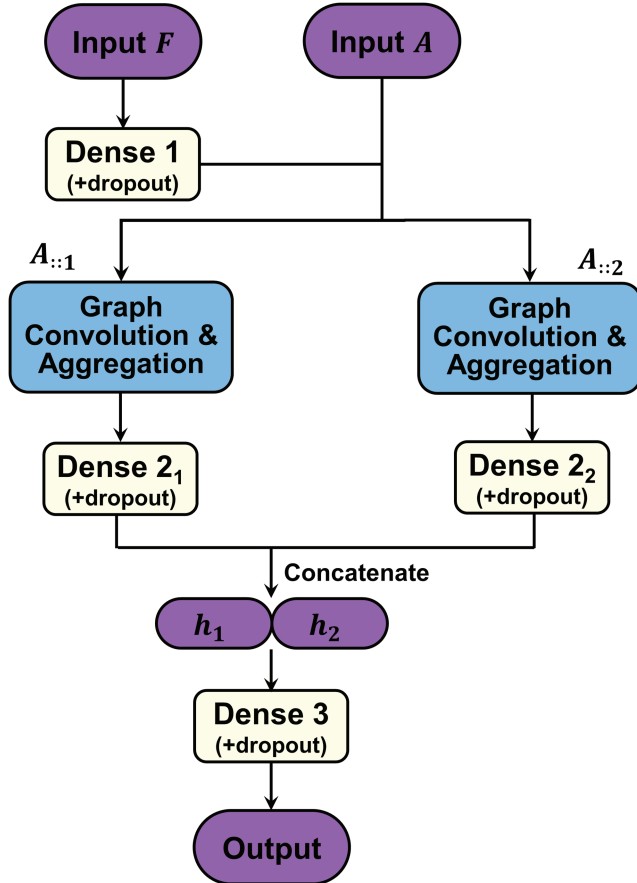

**Fig 3. A light graph-learning architecture adopted in this work.** The node feature matrix **F** and inter-molecular adjacency tensor **A** of a protein-ligand complex are the inputs, and the binding strength is the output. Main components of this architecture include graph convolution layers, node aggregation layers, dense (fully-connected) layers and dropout layers.

## Experiment and results

### Scoring performance of models

The aforementioned framework <u>scores</u> the binding strengths of protein-ligand complex structures through learning <u>A</u>tomic <u>G</u>raphs with <u>I</u>nter-<u>M</u>olecular <u>A</u>djacency (AGIMA-based scoring, abbreviated as *AGIMA-Score*). In an *AGIMA-Score* model, the binding area of a complex structure is treated as a graph, represented by a node-feature matrix $\mathbf{F} \in \mathbb{R}^{n \times m}$ and an adjacency tensor $\mathbf{A} \in \mathbb{R}^{n \times n \times K}$. Here the binding area is recognized as all the ligand atoms and the protein atoms within $4\mathring{A}$-distance of any ligand atom, referring to Son's work [20]. Three sets of node features, referring to *Pafnucy* ($\mathbf{F}^{18}$, $m = 18$) [14], *KDEEP* ($\mathbf{F}^8$, $m = 8$) [13] and *GraphBAR* [20] ($\mathbf{F}^{13}$, $m = 13$) respectively, were adopted to construct **F** (Table 1). $\mathbf{F}^{18}$ includes *generic physico-chemical* properties (e.g. atom types and partial charge) and *pharmacophoric* properties (e.g. aromaticity and hydrogen-bond membership) of atoms. $\mathbf{F}^{13}$ is a subset of $\mathbf{F}^{18}$ that excludes *pharmacophoric* properties. $\mathbf{F}^8$ focuses on *pharmacophoric* properties, with atomic charge and excluded volume considered.

**Table 1. Node-feature sets for building molecular graphs.** Three feature sets, with 18 features (from *Pafnucy*), 8 features (from *KDEEP*) and 13 features (from *GraphBAR*) respectively, were considered in this study. The names and data types of these features are listed.

| Feature set | Components | Data type |
|---|---|---|
| $\mathbf{F}^{18}$ | $f_1 \sim f_9$: one-hot encoding of atom types | |
| | (Boron, Carbon, Nitrogen, Oxygen, | |
| | Phosphorus, Sulphur, Selenium, | |
| | Halogen atom and Metal atom) | |
| | $f_{10}$: hybridization type | $f_1 \sim f_9$: binary |
| | $f_{11}$: number of heavy-atom neighbors | $f_{10} \sim f_{12}$: integer |
| | $f_{12}$: number of hetero-atom neighbors | $f_{13} \sim f_{17}$: binary |
| | $f_{13} \sim f_{17}$: pharmacophoric features (hydrophobicity, | $f_{18}$: float |
| | aromaticity, hydrogen-bond acceptor, | |
| | hydrogen-bond donor and ring membership) | |
| | $f_{18}$: partial charge | |
| $\mathbf{F}^8$ | $f_1 \sim f_5$: pharmacophoric features (hydrophobicity, | |
| | aromaticity, hydrogen-bond acceptor, | |
| | hydrogen-bond donor and metallicity) | $f_1 \sim f_5$: binary |
| | $f_6$: positive charge | $f_6 \sim f_8$: float |
| | $f_7$: negative charge | |
| | $f_8$: excluded volume | |
| $\mathbf{F}^{13}$ | $f_1 \sim f_{13}$: features in $\mathbf{F}^{18}$ except the | $f_1 \sim f_9$: binary |
| | pharmacophoric features ($f_{13} \sim f_{17}$) | $f_{10} \sim f_{12}$: integer |
| | | $f_{13}$: float |

When constructing the inter-molecular adjacency tensor **A** (Fig 2D), two distance ranges were selected for capturing multi-level protein-ligand interactions, as follows. As a pair of atoms having a distance $< 2\text{Å}$ are mostly connected by a covalent bond, we paid more attention to the atom pairs being $> 2\text{Å}$ apart for characterizing the inter-molecular interactions (non-covalent). Meanwhile, the binding area is recognized according to a pairwise atomic distance of $< 4\text{Å}$, we employed the two distance ranges, $(2\text{Å}, 3\text{Å})$ and $(3\text{Å}, 4\text{Å})$, in Eq 5 to build the adjacency tensor ($\mathbf{A} \in \mathbb{R}^{n \times n \times 2}$) in this work. Combining the three node-feature matrices ($\mathbf{F}^{18}$, $\mathbf{F}^8$ and $\mathbf{F}^{13}$) and the adjacency tensor (**A**) in the generation of molecular graphs, we constructed three *AGIMA-Score* models (*AGIMA-Score*$^{18}$, *AGIMA-Score*$^8$ and *AGIMA-Score*$^{13}$) based on the graph-learning architecture in Fig 3. To investigate whether a single distance range of $(2\text{Å}, 4\text{Å})$ can cover sufficient adjacency information, we built a single-matrix adjacency tensor ($\mathbf{A}_{SAM} \in \mathbb{R}^{n \times n \times 1}$) to pair up with the three node-feature matrices for each protein-ligand complex. This led to the construction of three new models (*AGIMA-Score*$_{SAM}^{18}$, *AGIMA-Score*$_{SAM}^8$ and *AGIMA-Score*$_{SAM}^{13}$) for comparison purpose. In addition, the non-redundant features ($\mathbf{F}^{21}$) from $\mathbf{F}^{18} \cup \mathbf{F}^8 \cup \mathbf{F}^{13}$ were collected and combined with the adjacency tensor **A** to build the *AGIMA-Score*$^{21}$ model. The dense layers each have a dimension of 128, and the number of epochs and batch size were tuned when constructing these models.

In order to evaluate the performance of these models comprehensively, several broadly-discussed, deep-learning scoring models were implemented as competing models. These include the Atom Convolutional Neural Network (*ACNN*) [27], *OnionNet* [18], *KDEEP* [13] and *GraphBAR* [20]. For *ACNN*, parameters including pooling filters, number of epochs and batch size were tuned in our work to reach the best model. Number of epochs and batch size were tuned for *OnionNet* and *KDEEP* (no data augmentation). As two similar graph-learning approaches, *GraphBAR* considers both intra- and inter-molecular contacts (Fig 2B) while *AGIMA-Score* focuses on only the inter-molecular contacts (Fig 2D) in the construction of molecular graphs. To make a fair comparison with *AGIMA-Score* models, we constructed

two *GraphBAR* models, *GraphBAR*$_{2AM}$ and *GraphBAR*$_{3AM}$, based on the architecture in Fig 3. *GraphBAR*$_{2AM}$ takes into account the intra-/inter-molecular contacts within $(0, 2Å)$ and those within $(2Å, 4Å)$ when building the adjacency tensor, which corresponds to the *AGIMA-Score*$_{SAM}$ models (considering inter-molecular contacts in $(2Å, 4Å)$). *GraphBAR*$_{3AM}$ adopts three distance ranges $((0, 2Å), (2Å, 3Å)$ and $(3Å, 4Å))$ to collect the intra-/inter-molecular adjacencies, corresponding to the *AGIMA-Score* models (considering inter-molecular contacts in $(2Å, 3Å)$ and $(3Å, 4Å)$). The number of epochs and batch size were treated as tuning parameters for the two *GraphBAR* models.

The *AGIMA-Score* and competing models were constructed based on the benchmark *PDBbind* database (https://www.pdbbind-plus.org.cn/). The *Refined Set* and *Core Set* in this database were employed for training and parameter tuning (validation). Each sample in these two sets is a protein-ligand complex structure (determined mostly by X-ray crystallography or NMR spectroscopy) with the experimentally resolved binding strength ($-logK_{d/i}$). These structural and binding-strength data are of high quality as they have gone through rigorous filtering processes [28,29]. To avoid potential train-validation contamination, each pair of complexes, one from the validation set and the other from the training set, needs to pass a similarity test. This test guarantees that the similarity of two protein sequences is below 0.3 or the similarity of two ligands is below 0.7, in each pair of complexes. Protein sequence similarities were generated using the *crossSetSim* function from the *protr* R library, with the default BLOSUM62 substitution matrix. Ligand similarities were calculated using the *cmp.similarity* function from the *ChemmineR* library, based on SMILES-transformed descriptors. The complexes against this rule were removed from the *Validation* set. Two sets from *CSAR* [30] were regarded as the final test sets, named *Test1* and *Test2*, in case of the over-optimistic results yielded from using the same-source data sets. The aforementioned similarity test was also performed on each test set vs. training set to prevent from potential train-test contamination. After this cleaning, the similarity statistics for pairwise complexes, with one complex from the training set and the other from the Validation, Test1, or Test2 set, are presented in Fig 4. Furthermore, the same protocol was adopted to ensure that there was no contamination among the *Validation*, *Test1* and *Test2* sets. Finally, the filtered *Training*, *Validation*, *Test1* and *Test2* sets consist of 5007, 195, 116 and 102

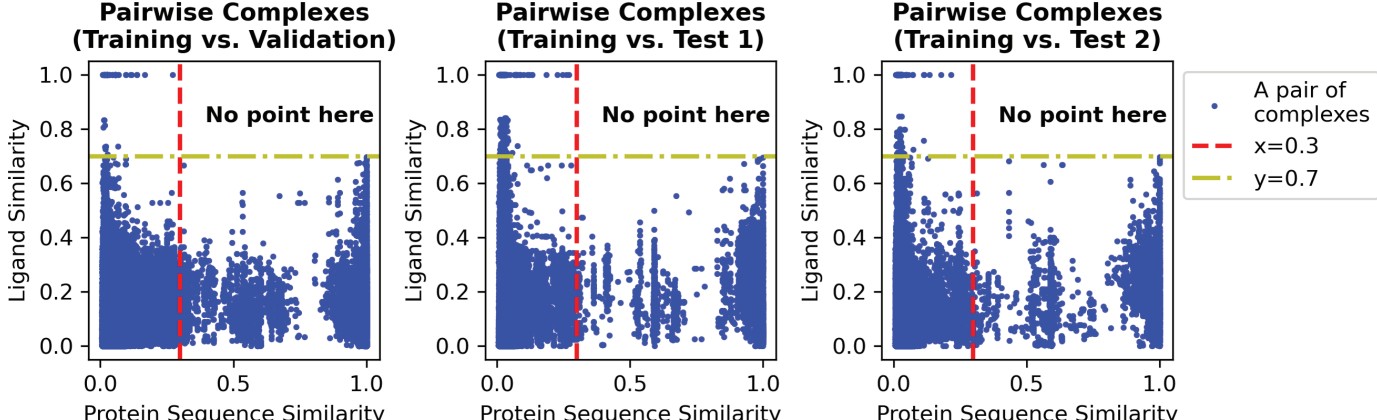

**Fig 4. Similarity test for each pair of complexes (*Training* vs. *Validation*, *Training* vs. *Test1*, and *Training* vs. *Test2*).** The horizontal axis stands for the similarity between the two protein sequences involved in a complex pair, and the vertical axis indicates the similarity between the two involved ligands. The red dotted line means a sequence similarity of 0.3 and the yellow line shows a ligand similarity of 0.7.

complex structures respectively. The lists of complexes for these sets can be found in *Zenodo* (https://zenodo.org/records/15023336).

The performance of each model was evaluated according to (1) the Pearson's Correlation (PC) between the experimental and predicted binding strengths of the complex structures and (2) the room-mean-square-error (RMSE) concerning those binding strengths. The evaluation results are now listed in Table 2.

An *ACNN* model is underfitted easily while a *KDEEP* model is often overfitted. Among the earlier models (*ACNN*, *OnionNet*, *KDEEP* and *GraphBAR*), *GraphBAR* outperforms the others in terms of *Test1*-PC, *Test1*-RMSE and *Test2*-RMSE, while *OnionNet* reaches the best PC on *Test2* set. Compared to these earlier models, the *AGIMA-Score* models perform well in average. *AGIMA-Score*[18] achieves the overall best performance because of the lowest *Test1*-RMSE, highest *Test2*-PC and lowest *Test2*-RMSE. *AGIMA-Score*[8] attains the best performance with respect to *Test1*-PC. Although using a single adjacency matrix ($AGIMA\text{-}Score_{SAM}$) eases the computations in graph learning, it often results in an underperformance in terms of PC and RMSE (compared to *AGIMA-Score*). It shows that using two adjacency matrices captures more information about the connections among atoms, such as the strong and weak hydrogen bonds with donor-acceptor distances of $2.2 \sim 2.5 \AA$ and $3.2 \sim 4.0 \AA$ separately [31]. The outperformance of *AGIMA-Score* over *GraphBAR* demonstrates the efficacy of learning the molecular graphs with inter-molecular adjacencies, rather than mixed intra- and inter-molecular adjacencies, in a scoring task. Overall, these results reveal the strong competitiveness of *AGIMA-Score* models in scoring the binding strength of a protein-ligand complex. A Docker container with the trained *AGIMA-Score*[18] model pre-installed is available for access on *Zenodo* (https://zenodo.org/records/15023336).

**Table 2. Scoring Performance Comparison.** The models were trained on *PDBbind Refined Set* (version *V2020*) with parameters tuned via the *Core Set* (version *V2020*), and tested on two sets from the *CSAR* source. State-of-the-art deep learning models (*ACNN*, *OnionNet*, *KDEEP* and *GraphBAR*) for scoring the protein-ligand complexes were realized, to comprehensively evaluate the proposed *AGIMA-Score* models. For *GraphBAR*, different graph adjacency schemes (2 or 3 adjacency matrices) were adopted for model construction. For *AGIMA-Score*, different node features (separately referring to *Pafnucy*, *KDEEP* and *GraphBAR*) and adjacency schemes (2 adjacency matrices or single adjacency matrix) were considered for model investigation. By default, 2 adjacency matrices (generated by intermolecular atomic contacts within $(2\AA, 3\AA)$ and those within $(3\AA, 4\AA)$) were adopted in the graph learning by *AGIMA-Score*. Best performance in terms of PC and RMSE were underlined for the state-of-the-art methods and the proposed *AGIMA-Score* models.

| Model | Training | | Validation | | Test1 | | Test2 | |
|---|---|---|---|---|---|---|---|---|
| | PC | RMSE | PC | RMSE | PC | RMSE | PC | RMSE |
| *ACNN* | 0.5225 | 1.7040 | 0.6586 | 1.7433 | 0.6079 | 1.8017 | 0.6064 | 1.7243 |
| *OnionNet* | 0.8745 | 1.0405 | 0.8663 | 1.3036 | 0.6547 | 1.7742 | **0.6543** | 1.6682 |
| *KDEEP* | 0.9909 | 0.2623 | 0.8310 | 1.2728 | 0.5339 | 1.9973 | 0.5963 | 1.7375 |
| $GraphBAR_{2AM}$ | 0.5906 | 1.5783 | 0.6736 | 1.7748 | **0.7192** | 1.6557 | 0.6540 | **1.6557** |
| $GraphBAR_{3AM}$ | 0.6184 | 1.5717 | 0.6922 | 1.6739 | 0.7178 | **1.6459** | 0.6450 | 1.6916 |
| *AGIMA-Score*[18]* | 0.6707 | 1.4638 | 0.7262 | 1.6949 | 0.7339 | **1.5806** | **0.6698** | **1.6279** |
| $AGIMA\text{-}Score_{SAM}^{18*}$ | 0.6540 | 1.4732 | 0.6978 | 1.6961 | 0.7154 | 1.6211 | 0.6042 | 1.7027 |
| *AGIMA-Score*[8]+ | 0.5533 | 1.6260 | 0.6976 | 1.7594 | **0.7414** | 1.6740 | 0.6466 | 1.6989 |
| $AGIMA\text{-}Score_{SAM}^{8+}$ | 0.5466 | 1.6313 | 0.7018 | 1.7351 | 0.7123 | 1.6860 | 0.6686 | 1.6509 |
| *AGIMA-Score*[13]@ | 0.6351 | 1.5396 | 0.7096 | 1.6912 | 0.7156 | 1.7102 | 0.6534 | 1.7456 |
| $AGIMA\text{-}Score_{SAM}^{13@}$ | 0.6023 | 1.5611 | 0.6634 | 1.7776 | 0.7177 | 1.6780 | 0.6398 | 1.7081 |
| *AGIMA-Score*[21]# | 0.6319 | 1.5180 | 0.6855 | 1.7478 | 0.7413 | 1.6327 | 0.6677 | 1.6549 |

*2AM* - 2 adjacency matrices (generated by atomic contacts within $(0, 2\AA)$ and those within $(2\AA, 4\AA)$) were used in the graph learning by *GraphBAR*.

*3AM* - 3 adjacency matrices (generated by atomic contacts within $(0, 2\AA)$, contacts within $(2\AA, 3\AA)$ and those within $(3\AA, 4\AA)$) were used in the graph learning by *GraphBAR*.

* Using 18 features for characterizing graph nodes (referring to *Pafnucy*).

+ Using 8 features for characterizing graph nodes (referring to *KDEEP*).

@ Using 13 features for characterizing graph nodes (referring to *GrahBAR*).

# Using 21 features for characterizing graph nodes (non-redundant features of those from *Pafnucy*, *KDEEP* and *GrahBAR*).

*SAM* - Single adjacency matrix (generated by intermolecular atomic contacts within $(2\AA, 4\AA)$) was adopted in the graph learning by *AGIMA-Score*.

## Screening performance of models

The scoring performance shows the capability of ranking a list of binding complexes or predicting the accurate binding strengths. Beyond that, the screening power is another indicator of interest for further evaluating the prediction models. As a more practical task, virtual screening aims to discover the potential binders for a target protein, in order to mitigate the burden of downstream biochemical experiments (Fig 5). Such a target protein often plays a key role in regulating the progression of some diseases, exemplified by the epidermal growth factor receptor (EGFR) protein that mediates the growth of non-small-cell lung cancer (NSCLC). Modeling the binding structure of each protein-ligand pair (*docking*) and scoring the binding strength based on this structure (*scoring*) are the two primary subtasks in virtual screening. Current state-of-the-art docking tools (e.g. *AUTODOCK* [32] and *Glide*[33]) can provide near-experimental binding structure for a pair of protein and ligand, while accurately scoring such a binding structure has long been a challenge. This work aims mainly at the scoring phase. Once we have the scored binding structures according to a model, *whether the true binders for the target protein can be highly ranked* is a main indicator of the screening capability of this model. *A model with both high scoring and high screening power is always the pursuit of the CADD community.*

To evaluate the screening power of *AGIMA-Score* models and their competitors, we selected a comparably large set from the *DUD-E* source (https://dude.docking.org). This set concerns the aforementioned EGFR and its potential ligands. A total of 36,273 ligands have been included in this set, with 832 actives (binders) and 35,441 decoys (non-binders) for the EGFR protein. Noteworthily, the models discussed above accept protein-ligand binding structures as inputs, but this *EGFR* set only contains the structures of the monomers (EGFR protein and ligands). Accordingly, we paired up the protein with each ligand into a binding structure by the well-acknowledged *AUTODOCK Vina* docking tool, before feeding the structures into the models. The best binding pose was retained for each protein-ligand pair, based on the default setting in *AUTODOCK Vina* and a reference structure (PDB:2RGP). The generated 36,273 EGFR-ligand binding structures were then fed into each model (*ACNN,*

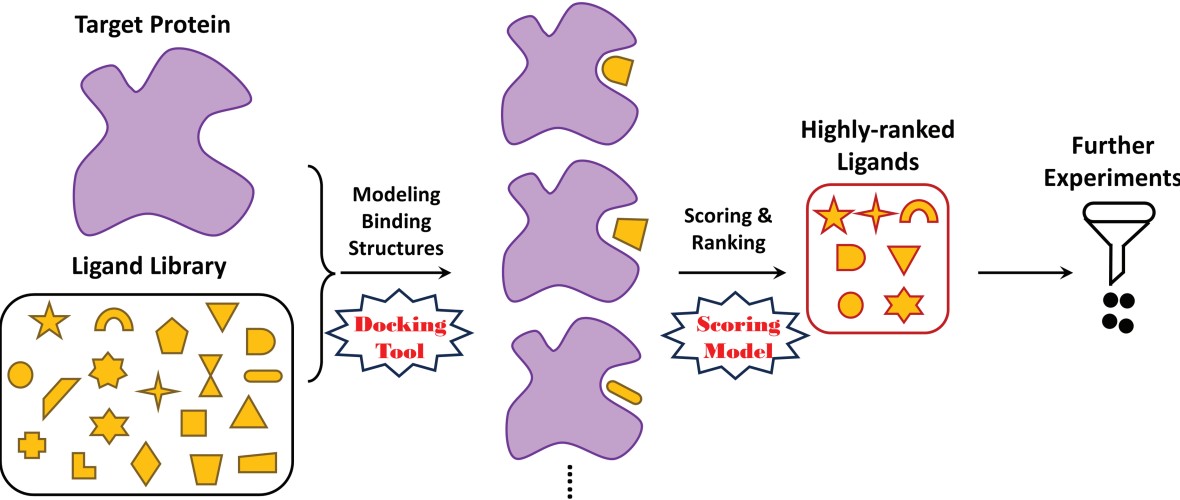

**Fig 5. A virtual screening task.** The task starts from a target protein and a big library of ligands, followed by the modeling of each protein-ligand binding structure (docking tool) and the scoring of the binding structures (scoring model). The highly-ranked ligands (according to the predicted scores) will be regarded as potential binders for further biochemical experiments.

*OnionNet, KDEEP, GraphBAR$_{2AM}$, GraphBAR$_{3AM}$, AGIMA-Score*[18]*, AGIMA-Score*[8] *and AGIMA-Score*[13]) to predict the binding strengths.

Enrichment factor (EF) is a widely-used index for evaluating the screening performance of a scoring model. It is defined as $EF^X = \dfrac{Y\%}{X\%}$, where $Y\%$ is the percentage of actives in the top $X\%$ ranked ligands. Meanwhile, the total decoy-to-active ratio ($r_{DTA}^{max}$) for this set is $\frac{35,441}{832} \approx$ 43, indicating a high imbalance between actives and decoys. To provide a more comprehensive evaluation, we composed a series of secondary sets according to varying $r_{DTA}$ values and assessed the corresponding EFs for each model on these sets. Aiming at a specific model **MDL**, an $r_{DTA}$ ($r_{DTA} = 2, \dots, r_{DTA}^{max}$) and an $X$ value, this procedure is described as follows.

- Keep all the 832 actives and randomly select $832 \times r_{DTA}$ decoys to constitute a set of size $832 \times (1 + r_{DTA})$.
- According to **MDL**, score and rank all the EGFR-ligand complexes in each of the set generated above, and calculate $EF_{r_{DTA}}^X$.
- Repeat the process for 10 times to derive the average EF ($\overline{EF_{r_{DTA}}^X}$).

The top $1 \sim 4\%$ ($X = 1 \sim 4$) ranked ligands were used for evaluating the screening performance of each model in Table 2, and the results are now exhibited in Fig 6.

Here, EF goes worse as $r_{DTA}$ or $X$ goes larger for all the models. However, *AGIMA-Score*[18] performs markedly better than the others. Encouragingly, *AGIMA-Score*[18] even achieves an EF of 26 for the top 1% ranked ligands, when all the decoys are included in the assessment.

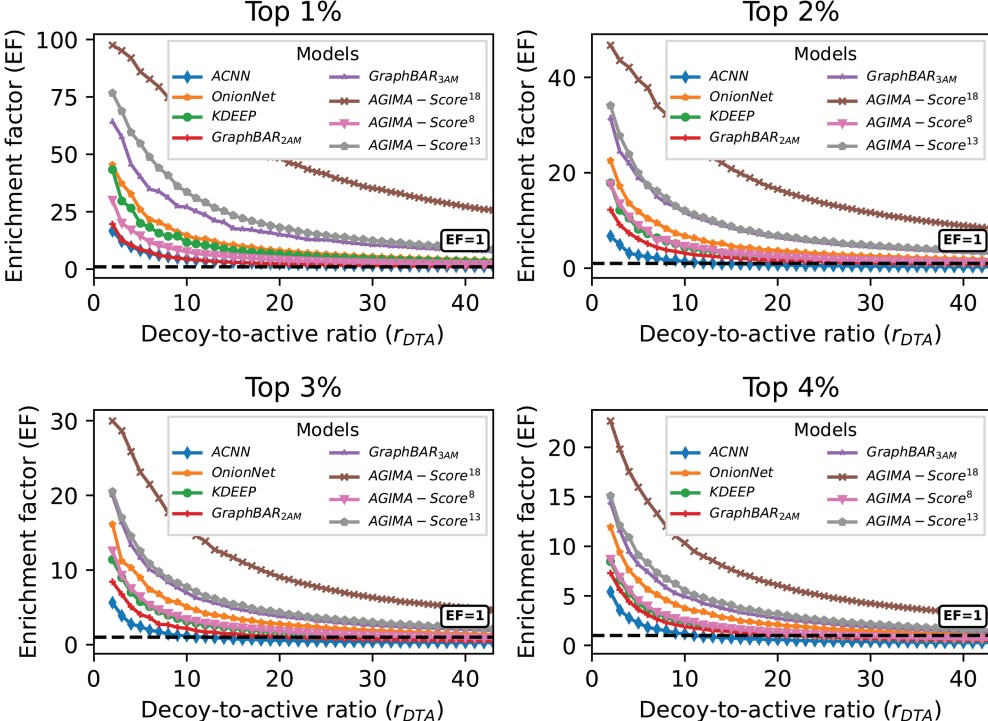

**Fig 6. The screening performance of each model on the *EGFR* set, with the top** $1 \sim 4\%$ **ranked ligands considered.** In each scenario, the enrichment factors (EFs) regarding various decoy-to-active ratios ($r_{DTA}$) were calculated for each model, and plotted in a line. The black dashed lines indicate EF = 1.

Similar results for the top $5 \sim 8\%$ ranked ligands are displayed in S1 Fig. Three more tasks, involving target proteins of HIV protease (HIVPR), ADAM17 Protease (ADA17) and tyrosine-protein kinase SRC (SRC), were considered. The *HIVPR* set covers 37,673 potential ligands (1,395 actives/36,278 decoys) for HIVPR protein. 30,956 (959 actives/29,997 decoys) and 35,790 (831 actives/34,959 decoys) ligands are included in the *ADA17* and *SRC* sets, respectively. The screening performance of the *AGIMA-Score* models and competitors on these three sets, in terms of the top $1\sim2\%$ ranked ligands, are displayed in Fig 7. For *HIVPR* set, *AGIMA-Score*[13] performs the best, with an EF of 29 for the top 1% ligands when involving all decoys. *AGIMA-Score*[8] outperforms the others for the *ADA17* set and *AGIMA-Score*[18] is the best performer for the *SRC* set, with EFs of 13 and 17 for the top 1% ligands (all ligands involved) respectively. The results for the top $3 \sim 4\%$ ranked ligands are presented in S2 Fig. Such results promote the *AGIMA-Score* models further.

## Discussion on model interpretability

Interpretations of deep learning models can build confidence in their predictions, therefore attracting more and more attention in recent years. Here, we discuss the ways to interpret *AGIMA-Score* models from `model level` and `post-hoc analysis`.

 **Model-level interpretation.** Due to the black-box nature of deep learning models, explaining the intrinsic structures of these models, which often concern millions or even

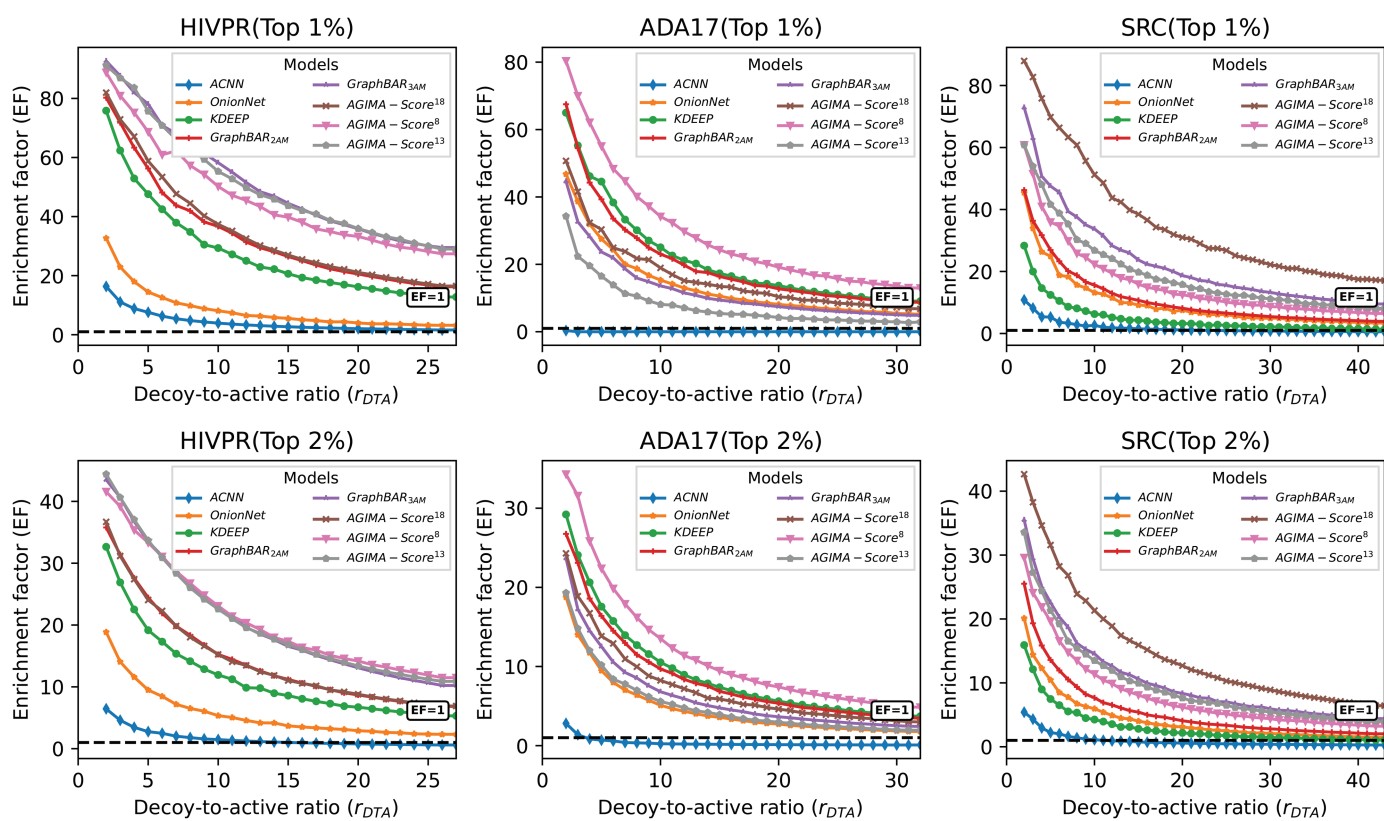

**Fig 7. The screening performances of *AGIMA-Score* and competing models on the *HIVPR*, *ADA17* and *SRC* sets, with the top $1 \sim 2\%$ ranked ligands considered.** In each scenario, the enrichment factors (EFs) regarding various decoy-to-active ratios ($r_{DTA}$) were calculated for each model, and plotted in a line. The black dashed lines indicate EF = 1.

more parameters, is quite difficult. In this regard, we focus mainly on the learning architecture (Fig 3) of *AGIMA-Score* models. This framework first transforms the original node features to an embedding space, and then considers multi-range, distance-dependent intermolecular interactions ($2\text{Å} < d_{ij} \leq 3\text{Å}$ and $3\text{Å} < d_{ij} \leq 4\text{Å}$) between a pair of protein and ligand. They imply important local interaction patterns between the two binding molecules. After a further feature-embedding transformation ($\mathbf{h}_1$ and $\mathbf{h}_2$), the framework gathers the information from those interaction patterns (by concatenation of $\mathbf{h}_1$ and $\mathbf{h}_2$) in the binding area. Then it maps the gathered information into the components of molecular binding strength or interaction energy using another hidden layer, leading to the final prediction of total binding strength. Hence, the framework can be partly explained in the perspective of molecular interaction energies.

**Post-hoc interpretation.**  After a model is constructed, investigating the roles of different features in the decision-making process and monitoring the correlations between some hidden features and the outputs are well-acknowledged strategies for decoding the model in a post-hoc way. Specifically, we employed the *masking-based feature importance assessment* and *principal component analysis (PCA) of key feature embeddings* in our work.

*Masking-based feature importance assessment.* To simplify the scenario, ascertaining the importance of each node feature in the decision-making process for a given *AGIMA-Score* model is the goal here. For such a model **MDL**, this assessment procedure is described as follows.

- Implement **MDL** on predicting the binding strengths of all the complexes in the validation set. Suppose the results are $PC_0$ and $RMSE_0$.
- Mask one node feature (*i*-th feature) at a time and re-implement **MDL** on the scoring task. Here, masking a feature means replacing the original features with 0s. As a result, a drop in PC ($\Delta PC_i$) and an increase in RMSE ($\Delta RMSE_i$) will be derived.
- Rank the node features in terms of PC drops (or RMSE increase), and those corresponding to a large PC drop (or RMSE increase) after being masked are more important in the decision-making process.

The assessment result for *AGIMA-Score*[18] model is now displayed in Fig 8. As shown here, certain pharmacophoric features (e.g. hydrophobicity, hybridization type and ring membership) weigh heavier importance than the atom types (e.g. Carbon, Nitrogen and Oxygen) in the perspective of either PC drop or RMSE increase. It verifies the important role of certain pharmacophoric properties in determining protein-ligand binding, as frequently applied in pharmacophore-based virtual screening [34]. However, other pharmacophoric properties, like hydrogen-bond donors, are of low interest in this scenario. The *AGIMA-Score*[8] model depends on a feature set that combines pharmacophoric properties, atomic charges and excluded volume (Fig 9). In this scenario, the excluded volume and atomic charges (positive or negative) stand out from the crowd of pharmacophoric features. The *AGIMA-Score*[13] model employs a simplified feature set of that from *AGIMA-Score*[18] (Fig 10). Here the partial charge, heavy-atom neighbors and hetero-atom neighbors dominate the PC drop, while atom types are more important in terms of RMSE increase. In summary of the importance plots, atom features such as certain pharmacophoric properties, atomic charges and connections play a vital role in revealing protein-ligand binding.

*PCA of key feature embeddings.* The feature embeddings ($\mathbf{h}_1$ and $\mathbf{h}_2$) in the last-but-two layer in Fig 3 were monitored in this study, because these hidden features stand for the important molecular interactions learned by an *AGIMA-Score* model. $\mathbf{h}_1$ and $\mathbf{h}_2$ correspond to the inter-molecular interactions in distance ranges $2\text{Å} \sim 3\text{Å}$ and $3\text{Å} \sim 4\text{Å}$ respectively. PCA

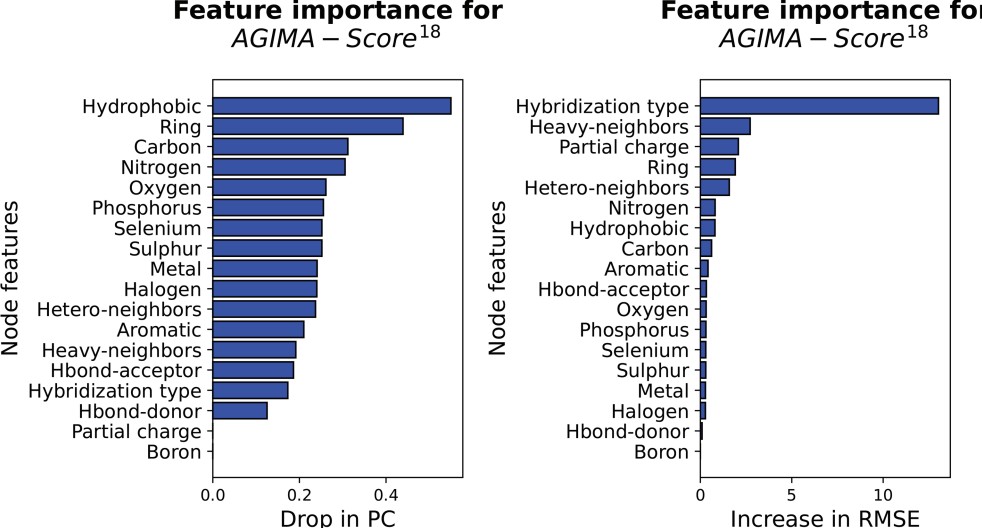

**Fig 8. Importance assessment of the node features involved in the *AGIMA-Score*[18] model.** The result was revealed by the masking-based performance drop on the validation set (*PDBbind Core Set*).

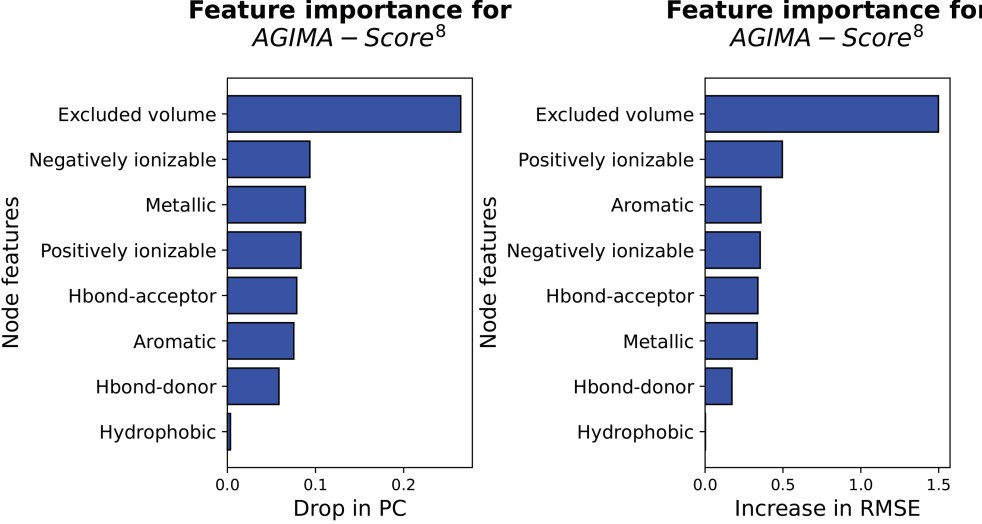

**Fig 9. Importance assessment of the node features involved in the *AGIMA-Score*[8] model.** The result was revealed by masking-based performance drop on the validation set (*PDBbind Core Set*).

was adopted to compress these embeddings, and explore their correlations with the molecular binding strength. In order for better visualization, the first principal component (PC1) for $\mathbf{h}_1$ and that for $\mathbf{h}_2$, of all the complexes in each set were extracted. Examining the correlations between such a PC (of $\mathbf{h}_1$ or $\mathbf{h}_2$) and the binding strength of a complex can provide useful insights into the logics of *AGIMA-Score* models. Taking *AGIMA-Score*[8] as an example, the *PC1 vs. binding strength* plots for the *Training, Validation, Test1* and *Test2* sets are now shown in Fig 11. The linear trend for $\mathbf{h}_1$-*PC1 vs. binding strength* and that for $\mathbf{h}_2$-*PC1 vs. binding strengths* were also captured in this figure, where a marked difference in the two

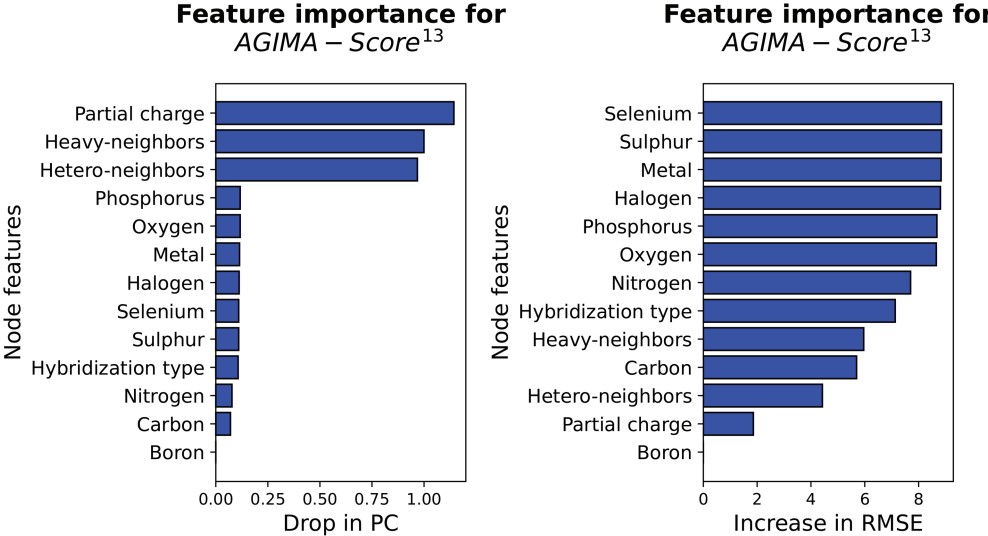

**Fig 10. Importance assessment of the node features involved in the *AGIMA-Score*[13] model.** The result was revealed by masking-based performance drop on the validation set (*PDBbind Core Set*).

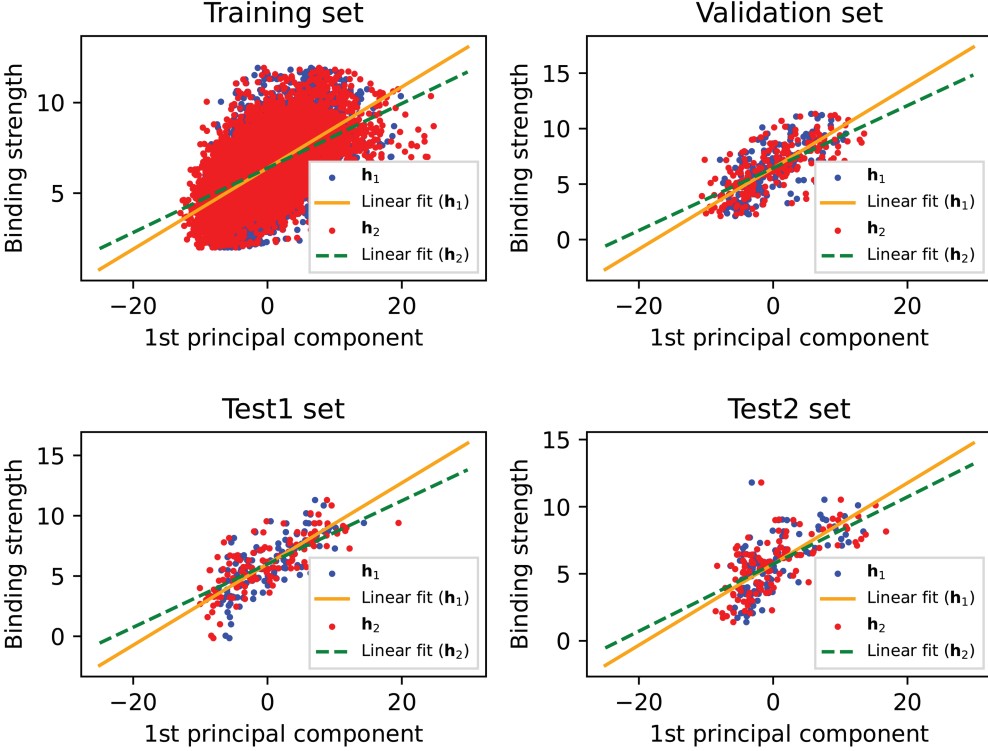

**Fig 11. Investigation of key feature embeddings in the *AGIMA-Score*[8] model.** The feature embeddings $\mathbf{h}_i$ ($i = 1, 2$) in the last-but-two layer of the model architecture were decoded by principal component analysis, and the first principal components of $\mathbf{h}_i$ were correlated with the binding strength via linear regression.

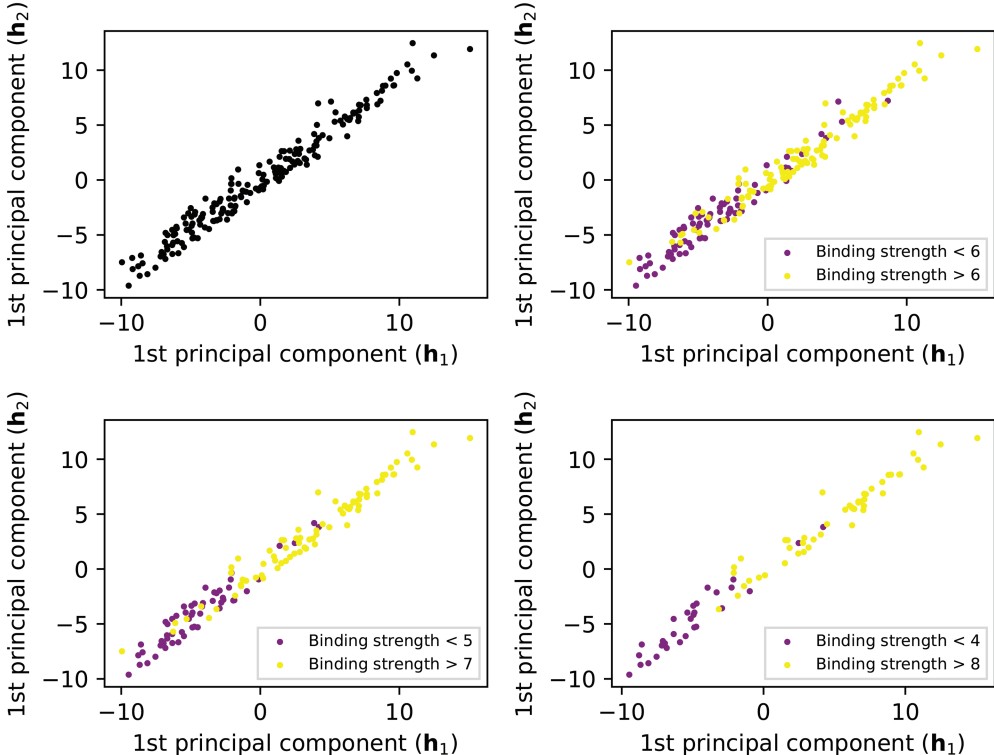

**Fig 12. Principal component plots of feature embeddings in the *AGIMA-Score*[8] model.** $h_1$-PC1 vs. $h_2$-PC1 plots for the validation set are shown. Different thresholds of binding strength were used to uncover the correlations between the PCs and the binding strength.

trendlines can be observed. This demonstrates two different types of interactions in the process of determining the protein-ligand binding strength.

Focusing on the *Validation* set, the scatter plots for the $h_1$-PC1 and $h_2$-PC1 of this model are displayed in Fig 12, where multiple thresholds for binding strength are also set to reveal the trends. It shows that higher values of interactions (represented by $h_1$-PC1 and $h_2$-PC1) normally lead to higher binding strengths. A similar analysis for *AGIMA-Score*[13] can be found in S3 and S4 Figs.

## Conclusion

The *AGIMA-Score* framework is introduced in this work. It describes a protein-ligand binding structure as an atomic-level graph, with only the inter-molecular interactions taken into consideration. Having a high computational efficiency, this framework places an absolute focus on the learning of the binding area of a protein-ligand complex. Depending on different sets of node features and a neat graph-learning architecture, a number of *AGIMA-Score* models were constructed. Such models perform well in the scoring of protein-ligand binding strengths and the screening of binders from non-binders for a target protein. At last, they can be explained reasonably from the model level, or in a post-hoc analysis. In the near future, our research will focus on exploring enriched sets of node features and developing more comprehensive approaches for model interpretability.

## Supporting information

**S1 Fig. The screening performance of each model on the *EGFR* set, with the top** $5 \sim 8\%$ **ranked ligands considered.** In each scenario, the enrichment factors (EFs) regarding various decoy-to-active ratios ($r_{DTA}$) were calculated for each model, and plotted in a line. The black dashed lines indicate EF = 1.
(EPS)

**S2 Fig. The screening performance of *AGIMA-Score* and competing models on the *HIVPR*, *ADA17* and *SRC* sets, with the top** $3 \sim 4\%$ **ranked ligands considered.** In each scenario, the enrichment factors (EFs) regarding various decoy-to-active ratios ($r_{DTA}$) were calculated for each model, and plotted in a line. The black dashed lines indicate EF = 1.
(EPS)

**S3 Fig. Investigation of key feature embeddings in the *AGIMA-Score*[13] model.** The feature embeddings $\mathbf{h}_i$ ($i = 1, 2$) in the last-but-two layer of the model architecture were decoded by principal component analysis, and the first principal components of $\mathbf{h}_i$ were correlated with the binding strength via linear regression.
(EPS)

**S4 Fig. Principal component plots of feature embeddings in the *AGIMA-Score*[13] model.** $\mathbf{h}_1$-PC1 vs. $\mathbf{h}_2$-PC1 plots for the validation set are shown. Different thresholds of binding strength were used to uncover the correlations between the PCs and the binding strength.
(EPS)

## Acknowledgments

No acknowledgments to declare.

## Author contributions

**Conceptualization:** Debby Dan Wang.

**Data curation:** Yuting Huang.

**Formal analysis:** Debby Dan Wang.

**Methodology:** Debby Dan Wang.

**Software:** Debby Dan Wang.

**Validation:** Yuting Huang.

**Writing – original draft:** Debby Dan Wang.

**Writing – review & editing:** Debby Dan Wang.

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
