## [Decision Letter · Decision Letter 0]

31 Jul 2024

Dear Dr. Wang,

Thank you very much for submitting your manuscript "Scoring Protein-ligand Binding Structures through Learning Atomic Graphs with Inter-Molecular Adjacency" for consideration at PLOS Computational Biology.

As with all papers reviewed by the journal, your manuscript was reviewed by members of the editorial board and by several independent reviewers. In light of the reviews (below this email), we would like to invite the resubmission of a significantly-revised version that takes into account the reviewers' comments.

We cannot make any decision about publication until we have seen the revised manuscript and your response to the reviewers' comments. Your revised manuscript is also likely to be sent to reviewers for further evaluation.

Sincerely,

Mohammad Sadegh Taghizadeh, Ph.D.

Academic Editor

PLOS Computational Biology

Nir Ben-Tal

Section Editor

PLOS Computational Biology

Reviewer's Responses to Questions

**Comments to the Authors:**

Reviewer #1: This paper aims to address the problem of predicting protein-ligand binding strength using a graph-based deep learning model. It represents interacting atoms as graph nodes and interactions as graph edges within two specific distance ranges. The study demonstrates competitive performance in scoring and screening tasks. Model interpretation reveals that features such as pharmacophoric properties play a vital role in complex scoring. Overall, this paper is well-written, and the analysis is comprehensive. However, I still have some concerns that need to be addressed:

Major:

1. The use of atomic graphs to represent protein-ligand interactions is well-justified. However, the choice of multi-level distance-dependent adjacency matrices (2-3 Å and 3-4 Å) needs further explanation. Why not use a single matrix (2-4 Å)? If there is a well-known reason for choosing multi-level adjacency matrices over a single one, please ignore it. I observe a strong correlation between h1 and h2 (Figure 9), indicating that h1 and h2 might convey similar features, or multicollinearity. Thus, I think multi-level adjacency might not be necessary. Please explain or discuss this.

2. Model Comparison: What are the key differences between the GraphBAR and AGIMA-Score 13 models? Both models appear to use 13 features and two adjacency matrices, yet AGIMA-Score 13 performs much better (e.g., in test set 1, 0.6463 vs. 0.6794). Please discuss the key model improvements and possible reasons.

Minor:

More features (e.g., 13 features) generally lead to better performance, and the paper also suggests it in both scoring and screening tasks. Naturally, I am wondering why the author didn' t combine those non-redundant features to train a single model. I have also seen language model embeddings used as additional features. The author might want to try it (no need for this paper). However, You might need to discuss feature selection for future model development based on your model interpretation.

Reviewer #2: The authors present AGIMA (Atomic Graphs with Inter-Molecular Adjacency), a graph-based method developed to predict binding affinities of protein-ligand complexes. The method also has screening capabilities, i.e., it can assign higher ranks to the true binders of the target protein.

Predicting the binding affinity of protein-ligand complexes is a very relevant question in structural biology. The field has experienced significant developments in predicting three-dimensional protein and protein complex structures in the last few years. However, binding affinity prediction is still an open question that needs to be solved. New computational methods are needed, and if successful, they will have a great impact on drug discovery and biomedical research.

AGIMA represents the protein-ligand complex as a graph where nodes are the atoms of the binding interface characterized by physico-chemical properties, and edges are defined by an adjacency tensor that handles two different cutoff distances to define inter-molecular contacts. A Chebnet Graph Neural Network (GNN) is used to learn from the constructed graphs.

Unlike other methods, AGIMA focuses only on inter-molecular interactions involving atoms of the binding interface. Additionally, it applies two different distance ranges (2-3 A and 3-4 A) to define these inter-molecular interactions.

The authors used the Refined (4856 cases) and Core (285 cases) Sets of the PDBbind database for training and validating the model, respectively. Two independent sets from the CSAR database, with 175 and 167 protein-ligand complexes, were used for testing. Additionally, a specific set of epidermal growth factor receptor (EGFR) protein-ligand structures from the DUD-E database was used to evaluate the models' screening capabilities.

AGIMA is compared with four other deep learning-based methods and shows a competitive performance.

The article is easy to read and well-structured. However, I suggest a review to correct some typos and the legibility of several sentences.

Additionally, I would propose the following suggestions for consideration:

* Main issues:

- One of the most critical elements when developing machine learning methods is having optimal datasets for training, validation and testing. Deep learning algorithms, in particular, are very demanding and usually require a large and diverse dataset during training. One main concern about this work is related to this aspect. The authors have used a dataset of 4856 instances to train the model. It is not clear that such a small dataset can be representative of the diversity of the protein-ligand complex conformational space and effectively used to train the model. An analysis of the dataset's characteristics would be helpful to address these concerns. What are the protein families represented in the datasets? What is the protein sequence variability within and between datasets? What is the variability of the ligands? The authors mention that there is no overlap between the training and test sets, but how this was enforced needs to be clarified. For example, a good approach to test the model's generalization capabilities would be to ensure that the proteins in the training and test sets do not share a sequence identity above a given threshold (e.g. 20-30%).

- A similar concern applies to the dataset used to evaluate the model's screening capabilities. The dataset only contains complexes for a single protein, the EGFR protein. How similar is EGFR to the proteins of the training set? In any case, I would suggest that a dataset containing more proteins should be used for this task to guarantee a good evaluation of the model's performance.

- Authors indicate the datasets used in their work. They also include the references to the original sources. As these sources might be subject to new versions, etc., I recommend storing the final datasets used in this work in a public repository lke Zenodo to guarantee the long-term availability of the data and reproducibility of the results. Ideally, the datasets and file structure defined at point "3. Data downloading and preprocessing" of the project Github repository (https://github.com/debbydanwang/DL-PLBAP) should be available at Zenodo or a public repository.

- The method requires the structure of the protein-ligand complex as input. This could be a limitation, as the number of available structures is very small compared with the number of available sequences. Have the authors considered assessing the method's performance using modelled protein-ligand complexes instead of experimental structures? It would be interesting to compare the results of the method using experimental or modelled structures as input. If performance with structural models was good, the method's applicability would increase.

- It would be interesting to include in the benchmark methods that do not require the structure of the complex but only the sequences as input.

- To improve clarity and understanding, I would suggest to expand the information provided in the table and figure captions of the paper so they are as self-explanatory as possible.

* Minor issues:

- The Results section includes key information about the datasets used in the work, the sets of features used to characterize the network nodes, and the method applied to evaluate the model's screening performance. I suggest providing this information in the Methods section.

- Could the authors indicate how much time does a prediction require on a standard machine (for example, seconds on a desktop PC or high-performance computing cluster)?

**Have the authors made all data and (if applicable) computational code underlying the findings in their manuscript fully available?**

Reviewer #1: Yes

Reviewer #2: Yes

PLOS authors have the option to publish the peer review history of their article (what does this mean?). If published, this will include your full peer review and any attached files.

Reviewer #1: No

Reviewer #2: No
---

## [Decision Letter · Decision Letter 1]

14 Sep 2024

Dear Dr. Wang,

Thank you very much for submitting your manuscript "Scoring Protein-ligand Binding Structures through Learning Atomic Graphs with Inter-Molecular Adjacency" for consideration at PLOS Computational Biology.

As with all papers reviewed by the journal, your manuscript was reviewed by members of the editorial board and by several independent reviewers. In light of the reviews (below this email), we would like to invite the resubmission of a significantly-revised version that takes into account the reviewers' comments.

We cannot make any decision about publication until we have seen the revised manuscript and your response to the reviewers' comments. Your revised manuscript is also likely to be sent to reviewers for further evaluation.

Sincerely,

Mohammad Sadegh Taghizadeh, Ph.D.

Academic Editor

PLOS Computational Biology

Nir Ben-Tal

Section Editor

PLOS Computational Biology

Reviewer's Responses to Questions

**Comments to the Authors:**

Reviewer #1: The authors have submitted a comprehensive revision, and I now consider this manuscript ready for publication. My primary concern in the initial submission was the justification for using multi-level distance-dependent adjacency matrices and incorporating all 21 non-redundant features in the final model. The revised approach demonstrates a significant improvement through the use of these adjacency matrices, which is clearly validated against other models. I am now fully satisfied with the authors' justification on this point.

Additionally, the responses to my other revision requests, as well as (in my view) those raised by the other reviewers, are clear, detailed, and comprehensive.

Reviewer #2: First of all, I would like to thank the authors for their responses to my comments.

While most of these responses have completely addressed my concerns, I still have a few comments that I believe need further clarification.

1. My initial question highlighted the importance of having a large and diverse training dataset. I suggested conducting an analysis of the dataset characteristics to evaluate these aspects. I also asked about the measures taken to ensure there was no overlap between the training and test sets.

In their response, the authors state that they have ensured no complexes are shared between the training and test sets. While it is true that the training and test sets do not share complexes with the same PDB ID, I believe that the training and test sets still contain models with different PDB IDs that represent the same complex. For example, the PDBs 2e2r and 2p7g, which are part of test1, represent the same complex (ESTROGEN-RELATED RECEPTOR GAMMA in complex with ligand bisphenol A) as PDB 6i63 in the training set. Similarly, PDB 1ax0 in test2 represents the same complex (ERYTHRINA CORALLODENDRON LECTIN in complex with ligand N-ACETYLGALACTOSAMINE) as PDB 3n35 in the training set. This means that the test sets used to evaluate the model contain complexes that the model has already seen during training, even if the experimental structures have different PDB IDs. To ensure the performance of the model has been evaluated correctly, all such instances should be identified and corrected. As the authors suggest in their response, a proper split would be one in which the complexes in the training and test sets do not share more than 30% sequence identity and ligand similarity does not exceed 70%.

The same precautions should be applied to the cases used to assess the potential of the models for virtual screening.

2. Regarding the diversity of the Refined set from which the training set was generated, the authors state that, during the generation of the Refined set, criteria were applied to ensure that the complexes do not share more than 30% sequence identity, to avoid redundancy, and that ligands do not exceed 70% similarity. However, the Refined set contains multiple instances of the same complex represented by different PDBs. This means that the number of unique complexes in the Refined dataset is lower than the 5,316 distinct PDB IDs it contains. To identify such issues, I suggested that the authors conduct an analysis of the diversity of the proteins and ligands in the datasets used. I believe this analysis remains relevant and would help assess the scope and limitations of the dataset used to train the model.

3. If the issues with the training and test sets described above are resolved and the model's performance remains strong, it would be very useful for the scientific community to have access to a Docker or Singularity container, or a server, with a trained AGIMA model available for use.

Thanks

**Have the authors made all data and (if applicable) computational code underlying the findings in their manuscript fully available?**

Reviewer #1: Yes

Reviewer #2: Yes

PLOS authors have the option to publish the peer review history of their article (what does this mean?). If published, this will include your full peer review and any attached files.

Reviewer #1: No

Reviewer #2: No
---

## [Decision Letter · Decision Letter 2]

22 Oct 2024

Dear Dr. Wang,

Thank you very much for submitting your manuscript "Scoring Protein-ligand Binding Structures through Learning Atomic Graphs with Inter-Molecular Adjacency" for consideration at PLOS Computational Biology.

As with all papers reviewed by the journal, your manuscript was reviewed by members of the editorial board and by several independent reviewers. In light of the reviews (below this email), we would like to invite the resubmission of a significantly-revised version that takes into account the reviewers' comments.

We cannot make any decision about publication until we have seen the revised manuscript and your response to the reviewers' comments. Your revised manuscript is also likely to be sent to reviewers for further evaluation.

Sincerely,

Mohammad Sadegh Taghizadeh, Ph.D.

Academic Editor

PLOS Computational Biology

Nir Ben-Tal

Section Editor

PLOS Computational Biology

Reviewer's Responses to Questions

**Comments to the Authors:**

Reviewer #2: I appreciate the clarifications in your response. However, I would like to address certain points that I believe require further attention.

1. Relying on visual or graphical assessments to determine structural similarity can be imprecise and subjective. I conducted an analysis of the structures using Cα-RMSD (C-alpha Root Mean Square Deviation) and obtained the following results:

The Cα-RMSD for the proteins in complexes 6i63 and 2p7g is 0.33 Å, and for 6i63 and 2e2r, it is 0.56 Å. Additionally, the RMSD of the ligands between 6i63 and 2p7g after superposing the proteins is 0.21 Å.

For complexes 3n35 and 1ax0, the Cα-RMSD is 0.28 Å, while the RMSD for their ligands is 0.54 Å.

These values are sufficiently low to consider the structures practically identical.

2. As you point out, proteins are flexible molecules that exist in a variety of conformations, often referred to as the "conformational ensemble." Experimental binding strength measurements reflect an average over this ensemble and are not associated with a single static structure. Therefore, I respectfully disagree with the statement that "each pair of complexes are structurally different, and such differences result in variations in the binding strengths." In my opinion, the complex pairs in question are structurally identical and most likely represent identical conformational ensembles. The differences in binding strengths are probably due to experimental errors, varying experimental conditions, or similar factors.

3. I might be repeating myself, but I believe a key issue that has been overlooked in the different versions of the paper is a thorough analysis of the degree of similarity between the training and test sets. Published studies have demonstrated that the similarity between sequences and structures in these sets can significantly impact the performance of machine learning and AI methods for predicting binding strength ([1], [2], [3]).

I consider important to include an in-depth analysis of the similarity of both proteins and ligands in the training and test sets. Additionally, studying how the model's accuracy varies depending on this similarity would provide valuable insights into the generalization power of the model.

4. I would also like to mention a point from my previous review regarding the number of unique complexes in the Refined dataset. I initially noted that the number of unique complexes is lower than the 5,316 distinct PDB IDs it contains. This is relevant because, if the model's generalization capacity is not high, it is important to have a wide and diverse training set. In response, you ran a protocol to identify valid pairs of complexes, defined as those with protein sequence similarity of less than 30% or ligand similarity of less than 70%. You also included the criterion of "different binding strengths." However, I would argue that this is not always a reliable criterion, as the binding strengths for the same protein-ligand complex can vary between different experiments due to differences in experimental conditions, measurement techniques, experimental errors, and other non-structural factors. You report finding 19 pairs of complexes that did not pass the filter. Could you please provide the raw computed results for pairwise protein similarity, ligand similarity, and binding strengths?

Thanks.

References:

1. Li Y, Yang J. Structural and Sequence Similarity Makes a Significant Impact on Machine-Learning-Based Scoring Functions for Protein–Ligand Interactions. J Chem Inf Model. 2017;57: 1007–1012. doi:10.1021/acs.jcim.7b00049

2. Su M, Feng G, Liu Z, Li Y, Wang R. Tapping on the Black Box: How Is the Scoring Power of a Machine-Learning Scoring Function Dependent on the Training Set? J Chem Inf Model. 2020;60: 1122–1136. doi:10.1021/acs.jcim.9b00714

3. Kramer C, Gedeck P. Leave-Cluster-Out Cross-Validation Is Appropriate for Scoring Functions Derived from Diverse Protein Data Sets. J Chem Inf Model. 2010;50: 1961–1969. doi:10.1021/ci100264e

**Have the authors made all data and (if applicable) computational code underlying the findings in their manuscript fully available?**

Reviewer #2: Yes

PLOS authors have the option to publish the peer review history of their article (what does this mean?). If published, this will include your full peer review and any attached files.

Reviewer #2: No
---

## [Decision Letter · Decision Letter 3]

20 Jan 2025

PCOMPBIOL-D-24-00595R3

Scoring Protein-ligand Binding Structures through Learning Atomic Graphs with Inter-Molecular Adjacency

PLOS Computational Biology

Dear Dr. Wang,

Thank you for submitting your manuscript to PLOS Computational Biology. After careful consideration, we feel that it has merit but does not fully meet PLOS Computational Biology's publication criteria as it currently stands. Therefore, we invite you to submit a revised version of the manuscript that addresses the points raised during the review process.

Please submit your revised manuscript within 60 days (May 17, 2025). If you will need more time than this to complete your revisions, please reply to this message or contact the journal office at ploscompbiol@plos.org. Please include the following items when submitting your revised manuscript:

We look forward to receiving your revised manuscript.

Kind regards,

Mohammad Sadegh Taghizadeh, Ph.D.

Academic Editor

PLOS Computational Biology

Nir Ben-Tal

Section Editor

PLOS Computational Biology

**Additional Editor Comments :**

Please carefully pay attention to the esteemed reviewer's comments and address them point-by-point.

**Reviewers' comments:**

Reviewer's Responses to Questions

Reviewer #2: In the new version of their work, the authors have taken steps to ensure that the training and test sets contain sufficiently distinct proteins and ligands. I believe these measures provide a more robust and reliable evaluation of the model. However, I have identified additional concerns regarding the validation and test sets:

The validation set and test1 set share the following complexes: 1vso, 2p4y, 2pog, 2qbq, 2qbr, 2vkm, 2vw5, 3bgz, 3e92, 3f3d.

The validation set and test2 set share the following complexes: 1gpk, 1h22, 1h23, 1nc1, 1nc3, 1ps3, 1qkt, 1r5y, 1s38, 1yc1, 1z95, 2fvd, 2hb1.

Keeping the validation and test sets completely disjoint is the standard best practice to ensure a robust and unbiased model performance estimate. Additionally, I believe the same protocol used to ensure dissimilarity between the training and test sets should be applied to guarantee that no similar proteins or ligands are shared:

- Between the validation set and the test sets

- Among the test sets themselves

Again, this will help maintain consistency and prevent potential data leakage, ensuring a more robust and reliable evaluation of the model.

In addition, I have the following suggestions and comments:

- It would be very useful for the scientific community to have access to a Docker or Singularity container, or a server, with trained AGIMA models available for use. This would facilitate reproducibility and allow other researchers to test and build upon your work without requiring training the model from scratch, etc.

- To further ensure reproducibility, please include the model’s predictions in the Zenodo repository together with the rest of the data.

- Please provide a description of the protocol used to ensure dissimilarity between proteins and ligands in the training and test datasets (software used, thresholds applied, etc). The method is described in the author's answer to the reviewers but has not been included in the paper.

- I have noticed several typographical issues that should be addressed. A thorough proofread and a minor revision of the text would improve clarity and readability. For example:

- Line 209 (Person's -> Pearson's)

- Line 66 ("However, designing task-oriented graph edges, particularly for those molecule-involved tasks, is always a challenging task.")

Thank you for considering these points.

**Have the authors made all data and (if applicable) computational code underlying the findings in their manuscript fully available?**

Reviewer #2: Yes

PLOS authors have the option to publish the peer review history of their article (what does this mean?). If published, this will include your full peer review and any attached files.

Reviewer #2: No

**Figure resubmission:**
---

## [Decision Letter · Decision Letter 4]

13 Mar 2025

PCOMPBIOL-D-24-00595R4

Scoring Protein-ligand Binding Structures through Learning Atomic Graphs with Inter-Molecular Adjacency

PLOS Computational Biology

Dear Dr. Wang,

Thank you for submitting your manuscript to PLOS Computational Biology. After careful consideration, we feel that it has merit but does not fully meet PLOS Computational Biology's publication criteria as it currently stands. Therefore, we invite you to submit a revised version of the manuscript that addresses the points raised during the review process.

Please submit your revised manuscript within 30 days (April 10, 2025; 11:59 PM). If you will need more time than this to complete your revisions, please reply to this message or contact the journal office at ploscompbiol@plos.org. Please include the following items when submitting your revised manuscript:

We look forward to receiving your revised manuscript.

Kind regards,

Mohammad Sadegh Taghizadeh, Ph.D.

Academic Editor

PLOS Computational Biology

Nir Ben-Tal

Section Editor

PLOS Computational Biology

**Reviewers' comments:**

Reviewer's Responses to Questions

Reviewer #2: I believe the latest updates introduced by the authors have improved their manuscript. However, I have a few minor remarks and suggestions that would further enhance the quality and reproducibility of their work:

Minor:

- It appears that the files test1.zip and test2.zip uploaded to the Zenodo repository contain identical data. Please verify and correct this to ensure each test set is accurately provided.

- While I appreciate that the authors have uploaded the model as a Keras file to Zenodo, I believe this alone is insufficient for optimal reproducibility and ease of use by the scientific community. If building a web server is not possible, I strongly suggest providing a Docker or Singularity container with the trained AGIMA model pre-installed, along with a clear use-case example demonstrating its application. This approach would greatly facilitate reproducibility and allow other researchers to test and build upon your work without requiring extensive setup or re-training of the model.

- I suggest to explicitly indicate in the manuscript that the trained model is available in the Zenodo repository, clearly specifying the format and instructions for usage.

Thank you for considering these points meant to enhance the quality, clarity, and reproducibility of your publication.

Best regards

**Have the authors made all data and (if applicable) computational code underlying the findings in their manuscript fully available?**

Reviewer #2: Yes

PLOS authors have the option to publish the peer review history of their article (what does this mean?). If published, this will include your full peer review and any attached files.

Reviewer #2: No

**Figure resubmission:**
---

## [Decision Letter · Decision Letter 5]

21 Apr 2025

Dear Dr. Wang,

We are pleased to inform you that your manuscript 'Scoring Protein-ligand Binding Structures through Learning Atomic Graphs with Inter-Molecular Adjacency' has been provisionally accepted for publication in PLOS Computational Biology.

Best regards,

Mohammad Sadegh Taghizadeh, Ph.D.

Academic Editor

PLOS Computational Biology

Nir Ben-Tal

Section Editor

PLOS Computational Biology

Reviewer's Responses to Questions

**Comments to the Authors:**

Reviewer #2: The authors have adequately addressed all the comments and concerns I previously raised. I believe the manuscript is now suitable for publication.

Best regards

**Have the authors made all data and (if applicable) computational code underlying the findings in their manuscript fully available?**

Reviewer #2: Yes

PLOS authors have the option to publish the peer review history of their article (what does this mean?). If published, this will include your full peer review and any attached files.

Reviewer #2: No

---

## [Editor Report · Acceptance letter]

PCOMPBIOL-D-24-00595R5

Scoring Protein-ligand Binding Structures through Learning Atomic Graphs with Inter-Molecular Adjacency

Dear Dr Wang,

I am pleased to inform you that your manuscript has been formally accepted for publication in PLOS Computational Biology. Your manuscript is now with our production department and you will be notified of the publication date in due course.

With kind regards,

Anita Estes
